# Generation of Multiple Arbovirus-like Particles Using a Rapid Recombinant Vaccinia Virus Expression Platform

**DOI:** 10.3390/pathogens11121505

**Published:** 2022-12-09

**Authors:** Yuxiang Wang, Anthony Griffiths, Douglas E. Brackney, Paulo H. Verardi

**Affiliations:** 1Center of Excellence for Vaccine Research, Department of Pathobiology and Veterinary Science, College of Agriculture, Health and Natural Resources, University of Connecticut, Storrs, CT 06269, USA; 2National Emerging Infectious Diseases Laboratories, Department of Microbiology, Boston University School of Medicine, Boston, MA 02118, USA; 3The Connecticut Agricultural Experiment Station, New Haven, CT 06511, USA

**Keywords:** recombinant vaccinia virus vectors, arboviruses, virus-like particles, Powassan virus, Heartland virus, severe fever with thrombocytopenia syndrome virus, Bourbon virus, Mayaro virus

## Abstract

As demonstrated by the 2015 Zika virus outbreak in the Americas, emerging and re-emerging arboviruses are public health threats that warrant research investment for the development of effective prophylactics and therapeutics. Many arboviral diseases are underreported, neglected, or of low prevalence, yet they all have the potential to cause outbreaks of local and international concern. Here, we show the production of virus-like particles (VLPs) using a rapid and efficient recombinant vaccinia virus (VACV) expression system for five tick- and mosquito-borne arboviruses: Powassan virus (POWV), Heartland virus (HRTV), severe fever with thrombocytopenia syndrome virus (SFTSV), Bourbon virus (BRBV) and Mayaro virus (MAYV). We detected the expression of arbovirus genes of interest by Western blot and observed the expression of VLPs that resemble native virions under transmission electron microscopy. We were also able to improve the secretion of POWV VLPs by modifying the signal sequence within the capsid gene. This study describes the use of a rapid VACV platform for the production and purification of arbovirus VLPs that can be used as subunit or vectored vaccines, and provides insights into the selection of arbovirus genes for VLP formation and genetic modifications to improve VLP secretion and yield.

## 1. Introduction

Virus-like particles (VLPs) are non-infectious nanoparticles that resemble native virions. They are based on viral structural proteins that self-assemble during natural infection or in expression systems. VLPs lack viral genetic materials and are exceptionally safe when used as subunit vaccines. The highly organized and repetitive architecture of these nanoparticles enables the presentation of conformational epitopes that can induce potent innate and adaptive immune responses [1,2,3]. Therefore, a number of VLP-based human vaccines are already licensed and many are being tested in preclinical and clinical trials [4].

In the past 50 years, arboviral diseases have been emerging or re-emerging at an alarming rate [5,6]. In addition to the well-known epidemics caused by Zika virus (ZIKV) [7], dengue virus (DENV) [8], West Nile virus (WNV) [5], and chikungunya virus (CHIKV) [9], many other arboviruses deserve our attention and action. Powassan virus (POWV) is a tick-borne flavivirus endemic to North America and the Russian Far East [10,11]. It can cause neurological disorders, in which case the fatality rate is about 12% [12]. Heartland virus (HRTV) and severe fever with thrombocytopenia syndrome virus (SFTSV, also known as *Dabie bandavirus* or *Huaiyangshan banyangvirus*) are two closely related tick-borne phleboviruses (*Phenuiviridae*) that are reported in the United States and East Asia, respectively [13,14,15,16]. They share similar clinical manifestations including fever, thrombocytopenia, and organ failure in fatal cases. Documented case fatality rates are up to 30% for HRTV and 18% for SFTSV [17,18]. Bourbon virus (BRBV) is a recently discovered tick-borne thogotovirus (*Orthomyxoviridae*) which led to the death of a previously healthy man from eastern Kansas, USA [19]. The case-patient was presented with fever, thrombocytopenia, reduced consciousness, and multi-organ failure 11 days post illness onset [19]. Mayaro virus (MAYV) is a mosquito-borne alphavirus (*Togaviridae*) that sporadically causes outbreaks in Central and South America [20,21,22,23,24]. From 1954 to 2019, a total 901 cases have been reported in Latin America and the Caribbean countries [25]. Signs and symptoms of a MAYV infection include fever, headache, and arthritis [26].

Many expression systems have been extensively studied and engineered for VLP production such as bacteria, yeast, plants, baculovirus-insect cell systems, mammalian cell lines, and cell-free systems. Vaccinia virus (VACV) is the orthopoxvirus used as a vaccine to eradicate smallpox. Recombinant VACV (rVACV) is still being actively used as a vector in vaccine research such as Ebola [27], HIV [28], and SARS-CoV-2 [29]. We have also shown the successful generation of ZIKV VLPs using a rVACV vector system [30]. Therefore, this study intends to apply the same strategy to generate VLPs for five different arboviruses.

At this point, no licensed vaccine or effective treatment is available for the aforementioned five arboviruses. As human activities keep overlapping with the expanding habitats of ticks and mosquitoes, particularly in the context of climate changes, the likelihood of outbreaks, epidemics, and pandemics that can result in fatal infections is on the rise worldwide. Here, we demonstrate the fast generation and purification of arbovirus VLPs by infecting Vero cells with replication-defective rVACV vectors rapidly generated in our laboratory [31], as a first step in the development of safe and effective vaccines to these public health threats.

## 2. Materials and Methods

### 2.1. Cells

African green monkey BS-C-1 (CCL-26), Vero (CCL-81), and human HeLa S3 (CCL-2.2) cells were purchased from the American Type Culture Collection (ATCC, Manassas, VA, USA). Cells were grown at 37 °C in 5% CO_2_ using Dulbecco’s modified Eagle medium (D-MEM; Thermo Fisher Scientific, Waltham, MA, USA) supplemented with 5–10% tetracycline-tested fetal bovine serum (FBS, Bio-Techne, Flowery Branch, GA, USA).

### 2.2. Viruses and Antibodies

The recombinant VACVs in this study were derived by sequential plaque purification [32] from a clone (9.2.4.8) of the L-variant of VACV (ATCC, VR-2035) strain Western Reserve (WR). Langat virus E monoclonal antibody was obtained from BEI Resources (National Institute of Allergy and Infectious Disease, National Institutes of Health, Bethesda, MD, USA; NR-40316). Mouse immune ascitic fluid against SFTSV, BRBV, and MAYV was acquired from the World Reference Center for Emerging Viruses and Arboviruses (University of Texas Medical Branch, Galveston, TX, USA). HRTV glycoprotein 1 (Gn) polyclonal antibody was purchased from Novus (Novus Biologicals, Centennial, CO, USA; NBP2-41228).

### 2.3. Arbovirus Gene Selection and Synthesis

All genes were codon-optimized for human expression and synthesized at GeneArt (Thermo Fisher Scientific, Waltham, MA, USA) or Genewiz (Azenta Life Sciences, Chelmsford, MA, USA) with engineered restriction endonuclease sites to facilitate subcloning into VACV transfer plasmids. The coding region is indicated in parentheses following each gene. The prM (163 amino acids) and E (498 amino acids) genes of POWV prototype lineage are based on GenBank accession NC_003687.1. The prM (163 amino acids) and E (498 amino acids) genes of POWV DTV lineage are based on GenBank accession AF311056.1. For all POWV constructs, a N-terminal methionine amino acid and a putative signal sequence (22 amino acids) within the C protein were included immediately upstream of prM. TargetP 1.1 software [33] (Technical University of Denmark) predicts the localization of POWV E protein and was used to guide the selection of optimal signal sequences. The membrane glycoprotein (1076 amino acids) and NP (245 amino acids) genes of HRTV are based on GenBank accession JX005845.1 and JX005843.1, respectively. The membrane glycoprotein (1073 amino acids) and NP (245 amino acids) genes of SFTSV are based on GenBank accession KP663744.1 and KP663745.1, respectively. The E (512 amino acids) and M (270 amino acids) genes of BRBV are based on GenBank accession KU708255.1 and KP657750.2, respectively. The 3′-ORF (1242 amino acids) of MAYV is based on GenBank accession KP842802.1. 

### 2.4. Generation of Replication-Inducible Recombinant Vaccinia Viruses (vINDs)

vINDs were generated as previously described [30]. Briefly, genes of interest were placed under the control of a constitutive VACV promoter in a transfer plasmid that contains a *tetR* repressor gene. The essential VACV gene D6R is controlled by the *tet*O_2_ operator immediately downstream of the natural D6R promoter [34]. When a second gene of interest is present, an additional synthetic VACV early/late promoter (P_sel_) was used to control its expression. A red fluorescent protein gene (DsRed) was inserted to facilitate sequential virus purification. The expression of DsRed is controlled by a synthetic VACV promoter or enabled by an internal ribosomal entry site (IRES) sequence from encephalomyocarditis virus, when two genes of interest are present. Transfer plasmids were mixed with FuGENE HD transfection reagent (Promega, Madison, WI, USA) then added to BS-C-1 cells infected with a *lac*-inducible parental virus expressing enhanced green fluorescent protein (EGFP) in 12-well plates. Cells were incubated at 37 °C for 48 h before collection. Serial plaque purification was completed by replacing the *lac* operon inducer isopropyl β-D-1-thiogalactopyranoside (IPTG) with the *tet* operator inducer doxycycline (DOX), as described in our recently developed EPPIC (Efficient Purification by Parental Inducer Constraint) approach [31]. Purified viruses were grown to a larger scale by infecting HeLaS3 cells. To confirm the presence of correct signal sequences, viral DNA was extracted (NucleoSpin Blood Mini kit; Macherey-Nagel, Bethlehem, PA, USA), PCR amplified (Q5 high-fidelity DNA polymerase; New England Biolabs, Ipswich, MA, USA) and sequenced. Primers are shown in Appendix A.

### 2.5. VLP Production and Purification

Vero cells in 100 mm dishes were infected with the recombinant vINDs at an MOI of 5 in the absence of tetracyclines, overlaid with 8 mL OptiPRO^TM^ serum-free medium (Thermo Fisher Scientific, Waltham, MA, USA), and incubated at 37 °C for 48 h before harvest. Cell culture supernatant was clarified at 7000× *g* for 20 min at 4 °C. A solution of 40% PEG-8000 was added to clarified supernatant to reach a final concentration of 8% before overnight incubation at 4 °C. PEG precipitated VLPs were concentrated using a SW 32 Ti rotor (Beckman Coulter, Indianapolis, IN, USA) at 9100 rpm for 30 min at 4 °C. VLP pellets were resuspended in TNE buffer (10 mM Tris, 120 mM NaCl, 1 mM EDTA, pH 8.0). Resuspended POWV, HRTV, and SFTSV VLPs obtained by PEG precipitation were directly used for electron microscopy imaging. BRBV and MAYV VLPs were further purified to obtain clearly visible TEM images as follows. Resuspended VLPs were loaded onto a 30% sucrose cushion and pelleted using a SW 55 Ti rotor (Beckman Coulter) at 35,200 rpm for 2 h at 4 °C. Pellet was resuspended in TNE buffer, added to a 20–50% linear sucrose gradient, and centrifuged using a SW 55 Ti rotor (Beckman Coulter) at 38,400 rpm for 4 h at 4 °C. Fractions containing VLPs were collected, combined, and concentrated using a SW 55 Ti rotor (Beckman Coulter) at 35,200 rpm for 2 h at 4 °C. Purified VLPs were resuspended in TNE buffer and stored at −80 °C.

### 2.6. Western Blot Analysis of VLPs

Mini-PROTEAN 4–20% TGX Stain-Free gels (BioRad, Hercules, CA, USA) and Trans-Blot Turbo Mini 0.2 µm Nitrocellulose Transfer Packs (BioRad) were used for SDS-PAGE and membrane transfer, respectively. For the quantitative comparison between signal sequences, each VLP sample was quantitated using a fluorometer (Qubit 4; Thermo Fisher Scientific, Waltham, MA, USA). This is to ensure the quantity of whole protein loaded to gels was the same across all samples. Membranes were blocked (5% non-fat milk and 0.05% Tween in PBS) overnight at 4 °C before a 2-h incubation in primary antibody solution. Membranes were washed (0.05% Tween in PBS) three times then incubated in goat-anti-mouse IgG conjugated with HRP (Thermo Fisher Scientific, Waltham, MA, USA) solution for 30 min. Membranes were washed an additional three times with PBS-Tween and once with PBS before adding substrate (Clarity Western ECL Substrate; BioRad) for 5–8 min. A digital imager (ChemiDoc MP; BioRad) was used to develop images.

### 2.7. Negative Staining and Transmission Electron Microscopy (TEM) of VLPs

Purified VLP samples were fixed with 2% glutaraldehyde (Electron Microscopy Sciences, Hatfield, PA, USA) for 15 min before electron microscopy. Fixed VLP samples (3 µL) were placed on plasma-cleaned carbon-coated copper grids (Electron Microscopy Sciences) and incubated for 2 min. Grids were stained by 0.5% uranyl acetate before imaging with a FEI Tecnai 12 G2 Spirit BioTWIN transmission electron microscope at the University of Connecticut Biosciences Electron Microscopy Laboratory. VLP size was determined using the ImageJ [35] software with about 20 VLPs. 

## 3. Results

### 3.1. Successful Generation of VLPs for Five Arboviruses with the EPPIC VACV Platform

We first rapidly constructed vINDs that carry genes of interest from each of the five arboviruses using the EPPIC platform for the rapid generation of rVACVs (Figure 1). This platform allows rVACVs of interest to be generated and purified within a week.

The schematic arrangement of genes in each construct is depicted in Figure 2A. In the case of VLPs assembled from a single open reading frame (vIND-POWV and vIND-MAYV), expression was controlled by a strong synthetic early/late VACV promoter P_E/L_ [36]. When two open reading frames were expressed (vIND-HRTV, vIND-SFTSV, and vIND-BRBV), one was controlled by P_E/L_ and another by a synthetic VACV early/late promoter P_sel_ [37]. The cassettes were placed between VACV D5R and D6R genes. The resultant vINDs only replicate in the presence of tetracycline antibiotics because the expression of essential gene D6R is regulated by *tet*R repressor gene and *tet*O_2_ operator [34]. More specifically, the TetR protein (produced under one of the back-to-back P_E/L_ promoters) binds to its operator sequence in the absence of tetracyclines, blocking the expression of the essential VACV D6R gene, and consequently arresting viral replication. To note, the absence of tetracyclines prevents vINDs from producing infectious progeny, but not the expression of heterologous genes of interest [34,38]. A fluorescent protein (DsRed) facilitated the rapid plaque purification of the vINDs and its expression was either under the control of P_sel_ or an internal ribosomal entry site (IRES) [39], as shown in Figure 2A.

We selected a combination of genes that would most likely lead to VLP formation for each of the five arboviruses. POWV has a 10,839-nucleotide genome of positive sense single-stranded RNA encoding seven nonstructural proteins and three structural proteins (the capsid protein C, the precursor membrane glycoprotein prM, and the envelope protein E) [40]. In addition to E protein, prM protein is essential during the maturation of flaviviruses [41]. Therefore, both the prM gene and the E gene are present in our constructs to ensure correct formation of POWV VLPs. We also included the signal sequence within the C gene immediately prior to the prM gene with the intention to efficiently translocate the protein to the endoplasmic reticulum (ER). MAYV has a 11.4-kb positive sense single-stranded RNA genome that encodes four nonstructural proteins and five structural proteins [42]. We expressed the entire 3′-ORF that contains the five structural genes (C, E3, E2, 6K, and E1), based on research of another alphavirus, CHIKV [43,44,45,46,47,48], since at the point of designing our study, there were no reported strategies to generate MAYV VLPs. HRTV and SFTSV are both phleboviruses (Phenuiviridae) and share similar genomic composition. The negative-stranded RNA genome has three segments: a 6.4-kb L segment encoding a RNA polymerase, a 3.4-kb M segment encoding glycoproteins Gn/Gc, and a 1.7-kb S segment encoding a nucleoprotein [49]. The expression of the Gn/Gc gene alone from some Bunyavirales were reported to form VLPs [50]. However, several minigenome studies constructing infectious virus-like particles [51,52,53] also included the nucleoprotein (NP) gene. Therefore, to ensure successful formation of HRTV and SFTSV VLPs, we decided to incorporate their NP genes as well. BRBV has a negative sense RNA genome composed of six segments that encode a polymerase, an envelope (E) glycoprotein, a matrix (M) protein, and a nucleoprotein [54]. It has been hitherto poorly investigated, and therefore we decided to include both the E (segment 4) and M (segment 6) genes to ensure the assembly of VLPs in reference to studies on other thogotoviruses [55,56]. 

We next investigated the expression of the viral genes by each vIND construct that was generated. We infected Vero cells, precipitated VLPs from culture supernatants, and performed Western blots to detect the expression of each gene (Figure 2B). Anti-Langat E antibody has been shown to cross-react well with POWV envelope protein [57,58,59]. It detected a major band of approximately 54 kDa for both prototype and DTV lineages. A minor band slightly larger than 54 kDa was occasionally seen (likely the M-E complex). HRTV glycoprotein 1 (Gn) antibody recognized a band slightly lower than 50 kDa, close to its theoretical molecular weight. The Gn, Gc, and N proteins of SFTSV have a molecular weight close to 58 kDa, 58 kDa, and 27 kDa, respectively. This is correctly reflected by the bands recognized by polyclonal antibodies, although a non-specific band around 58kDa was also obeserved in the control lane. Anti-BRBV polyclonal antibody detected a larger band of ~55 kDa representing the envelope protein and a smaller band of ~30 kDa representing the matrix protein (considerably fainter bands in the control lane are suggestive of spillover). Anti-MAYV polyclonal antibody bound to a ~50 kDa band which corresponds to the expected size of MAYV E1 and E2 proteins. The expected size of all proteins was verified by molecular weight computational tools [60].

To image POWV, HRTV, and SFTSV VLPs under TEM, PEG precipitation was sufficient. POWV VLPs had an average diameter of 58 nm (Figure 3A,B), similar to the native virion [61]. The native enveloped particles of HRTV average around 86 nm in diameter [15]. We observed VLPs averaging 68 nm (Figure 3C,D). MAYV has a diameter of 70 nm [62], identical to the average size of VLPs seen in this study (Figure 3E,F). SFTSV virions range from 80 to 100 nm [13], well resembling the 93-nm VLPs we observed (Figure 3G,H). To image VLPs of BRBV and MAYV under TEM, we further purified PEG-precipitated VLPs via sucrose cushion followed by sucrose gradient. BRBV was reported to have two distinct shapes: ~1-µm filamentous virions and 100- to 130-nm spherical particles [19]. This is in agreement with our observation that BRBV VLPs also had both filamentous and spherical morphology (Figure 3I–L). However, the size were around 437 nm (filamentous) and 49 nm (spherical), respectively, significantly smaller than native virions.

### 3.2. Improvement in the Yield of POWV VLPs

Based on our prior work with ZIKV [30], we next decided to attempt to increase the yield of POWV VLPs by modifying the signal sequence for the prM-E genes. A typical signal sequence encodes a short peptide that contains: (1) a positively-charged N-terminus, (2) a hydrophobic region, and (3) a polar uncharged C-terminus [63]. Upon analyzing the natural signal sequence within the POWV DTV lineage, we realized that the negatively-charged aspartic acid residue in the N-terminal region might impair the translocation of protein to the ER (Figure 4A). Therefore, we replaced it with a hydrophobic tryptophan residue or a positively-charged lysine residue. We also tested the signal sequence derived from another flavivirus, Japanese encephalitis virus (JEV), that has been used successfully to promote secretion of proteins [64]. We then used the Target P 1.1 software [33] to predict the localization of prM-E protein after modification. The higher the prediction score, the more likely the protein is targeted to the secretory pathway (Table 1). We tested two modifications with a lower (D6K, 0.850 < 0.886) or equal (JEV, 0.886 = 0.886) score compared to the natural signal sequence counterpart, because they were used successfully in a study with ZIKV [30], another flavivirus. 

Compared to the natural signal sequence (D at position 6), the D6K variant showed no difference while the JEV variant showed a significant decrease in the amount of VLPs collected from cell culture supernatants (Figure 4B). The D6W mutation, however, increased VLP secretion (Figure 4B), similar to what we observed in our published ZIKV study [30] (Appendix A). Thus, the D6W mutation was beneficial while the replacement with the JEV natural signal sequence was deleterious (Figure 4B and Table 1).

## 4. Discussion

VACV has many advantages that make it a great viral vector for vaccine development, even four decades after eradication of smallpox. VACV is stable within a wide range of temperatures, readily propagable, not oncogenic, and most importantly, capable of inducing potent humoral as well as cell-mediated immune responses [65]. Our *tet*-inducible vINDs contain a built-in safety mechanism that eliminates the production of progeny viruses in the absence of tetracycline inducers, while allowing high levels of expression of inserted genes of interest even in the absence of inducers [34,38]. vINDs can be easily propagated in the presence of tetracyclines, and later administered as vaccines in the absence of tetracyclines. This feature addresses concerns of possible complications from vaccination, particularly in individuals with conditions such as atopic dermatitis, cardiac disease, and immunosuppression. Additionally, our EPPIC system enables rapid generation of vINDs to test the expression of single or multiple genes in as little as a week [31]. Therefore, the vINDs generated in this study could serve as vaccine candidates for the five arboviruses tested, with the added benefit that they also produced and secreted VLPs.

Due to the limited availability of primary antibodies, especially for emerging or neglected arboviruses, we were not able to detect each protein component individually using Western blots. The anti-HRTV mouse immune ascitic fluid (MIAF) failed to show any protein expression (data not shown) and of the two polyclonal antibodies (G1 and G2) we purchased from Novus, only G1 antibody showed reactivity. SFTSV construct should express three proteins: 58 kDa glycoproteins Gn and Gc, and a 27 kDa nucleoprotein. The MIAF against SFTSV recognized highly expressed nucleoprotein but could not distinguish Gn from Gc, especially since a band of similar size was also detected in the negative control (Appendix A). Like other alphaviruses, MAYV VLPs should have a surface structure of heterodimeric E2/E1 spikes [66,67]. With the MIAF, we detected a 50 kDa band which well represents the size of E1 and E2 protein. However, we could not specifically attribute this band to E1 or E2, nor could we observe the expression of other protein components (C, E3, and 6K) contained within our MAYV construct. Therefore, the development of primary antibodies for various arboviruses to aid in protein expression studies, VLP confirmation, and vaccine and pathogenesis studies is of high importance.

We observed minor differences between the size of some VLPs in this study and native virions as described in prior studies. The HRTV VLPs obtained have an averaged diameter of 68 nm, smaller than the reported diameter of virions for Bunyaviruses (90 to 100 nm) [68]. We calculated a size of 49 nm and 437 nm, respectively, for the spherical and filamentous BRBV VLPs, both smaller than the 1 µm (filamentous) and 100–130 nm (spherical) size of native BRBV virions. This could be due to inaccuracies when performing the measurements and the lack of uniformity in VLP sizes. More importantly, native virions are internally associated with nucleic acids and nonstructural proteins such as polymerases, while VLPs are essentially empty. This difference in composition could very likely result in the smaller size observed for the VLPs.

Our attempt to improve the VLP yield by modifying the signal sequence for POWV prM-E proteins showed similar results to what we discovered in a previous ZIKV vaccine study [30]. We observed improvement for POWV DTV lineage when the aspartic acid was replaced with a tryptophan. However, replacement with a lysine or the entire signal sequence of JEV did not lead to such an increase. The fact that modifying negatively charged amino acid residues within the N-terminal region of a signal sequence does not always lead to an increase in expression may be a result of other factors involved in the process of POWV VLP biosynthesis and secretion. Additionally, the JEV natural signal sequence may not always succeed in improving secretion. Overall, to improve VLP secretion via genetic engineering, additional approaches need to be explored such as better characterization of signal sequences for a more appropriate design, alteration of the transmembrane domain of the envelope protein to facilitate VLP releasing [69,70], and optimization of cleavage sites to enhance VLP maturation [71]. Our discovery offers an incentive to further study signal sequences and our approach in this study may provide more significant improvement of VLP secretion in the future if combined with other sequence modifications.

Arboviral diseases continue to threaten public health worldwide. With global climatic changes, arthropod vectors have the potential to expand both in number and distribution. For example, the Asian long-horned tick (*Haemaphysalis longicornis*), native to eastern Asia, is now rapidly spreading in the United States [72] and it is capable of transmitting several pathogens including SFTSV [13,73,74]. Despite extensive research on a few arboviruses, the majority are still poorly understood, and no licensed vaccines or specific antiviral treatments are available. The five arboviruses in this study (POWV, HRTV, SFTSV, BRBV, and MAY) can lead to severe diseases and mortality, and some have caused outbreaks, while some have yet to become prevalent and threating. However, we should be prepared for epidemics and possible pandemics, particularly in light of the lessons learned during the 2015 ZIKV outbreak. 

VLPs have been widely used in vaccine development. In fact, the human papillomavirus (HPV) and several human hepatitis B virus (HBV) vaccines on the market are VLP-formulated [4]. Because of its exceptional safety profile and similarity to native viruses, VLPs are excellent choices for developing vaccine candidates for arboviruses. POWV VLPs [57] produced in HEK 293T cells and MAYV VLPs [75] generated using a recombinant adenovirus have been reported. Here, we proved the feasibility of making VLPs in a different system and provided an approach to improving the yield of VLPs for POWV. While HRTV and SFTSV have been studied in multiple minireplicon assays [52,76], their VLP images have not been published. In addition, to the best of our knowledge, we are the first to show BRBV VLPs. In this study, we acquired high-resolution TEM images of VLPs expressed by VACV and we hope that our efforts will advance the development of VLP-based vaccines for tick- and mosquito-borne viruses. Indeed, in our previous study we demonstrated that vIND-ZIKV, a vaccine candidate for ZIKV, induced potent antibody and cell-mediated immune responses in mice that were protected from weight loss and viremia after challenge [30]. This finding indicates that VLPs produced by vINDs are sufficiently expressed in vivo in the absence of inducers such as tetracyclines to elicit protective immune responses.

## Figures and Tables

**Figure 1 pathogens-11-01505-f001:**
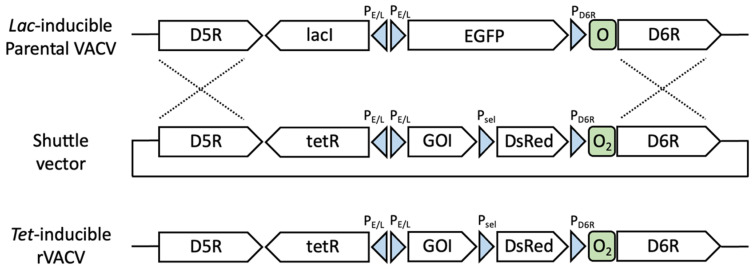
Recombinant VACVs were generated by homologous recombination using the EPPIC VACV plaftorm. A *lac*-inducible parental VACV (**top** panel) contains the D6R promoter (P_D6R_) followed by the *lac*O operator sequence (O) and the *lac*I and EGFP genes under back-to-back synthetic P_E/L_ promoters. The DNA shuttle vector (**middle** panel) contains the D6R gene under a *tet* operator (O_2_)-controlled P_D6R_ promoter, *tet*R and an arbovirus gene of interest (GOI) under back-to-back P_E/L_ promoters, and the DsRed gene under a synthetic early/late promoter (P_sel_) flanked by regions homologous to the D5R and D6R genes. Homologous recombination (dashed lines) between the shuttle vector (**middle** panel) and the *lac*-inducible parental VACV (**top** panel) generates *tet-*inducible rVACVs expressing DsRed (**bottom** panel), which is purified away from the parental virus within a week by switching inducers (IPTG to DOX). Schematics not drawn to scale.

**Figure 2 pathogens-11-01505-f002:**
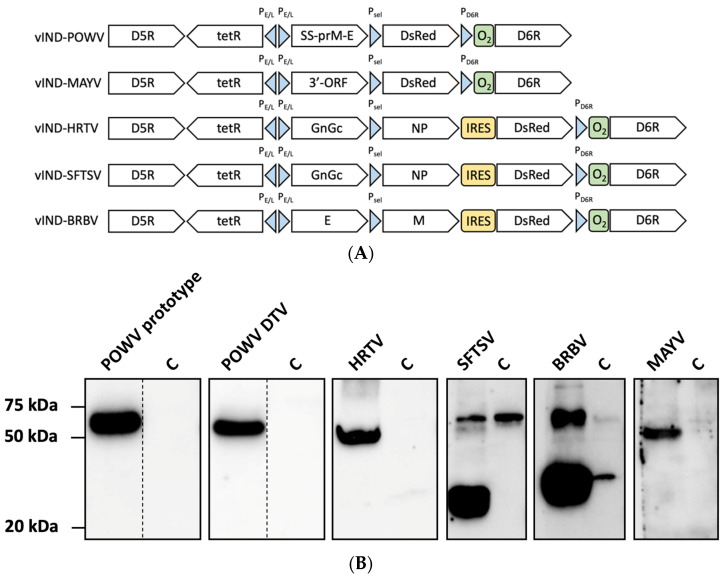
vIND constructs express and secrete heterologous arboviral proteins of interest into cell culture supernatant. (**A**) Schematic representation of vIND constructs. All vIND constructs contained genes of interest placed in the D5R-D6R locus of VACV. P_E/L_: synthetic early/late VACV promoter. P_sel_: synthetic early/late VACV promoter. P_D6R_: natural promoter of VACV D6R gene. *tet*R: *tet* repressor gene. O_2_: *tet* operator sequence. IRES: internal ribosomal entry site sequence from encephalomyocarditis virus. DsRed: red fluorescent protein gene. SS-prM-E: POWV signal sequence, prM, and E gene within a single open reading frame. 3′-ORF: the 3′-end open reading frame of MAYV. GnGc: glycoprotein gene Gn and Gc of HRTV or SFTSV. NP: nucleoprotein gene of HRTV or SFTSV. E: BRBV envelope gene. M: BRBV matrix gene. (**B**) Detection of protein expression in Western blots. Protein samples (VLPs) were PEG precipitated from culture supernatant of Vero cells and detected after SDS-PAGE with specific secondary antibodies. A negative control (C) for each sample is also shown immediately to the right. A dashed line indicates that lanes were not continuous, and full-length blots with negative controls can be found in Appendix A.

**Figure 3 pathogens-11-01505-f003:**
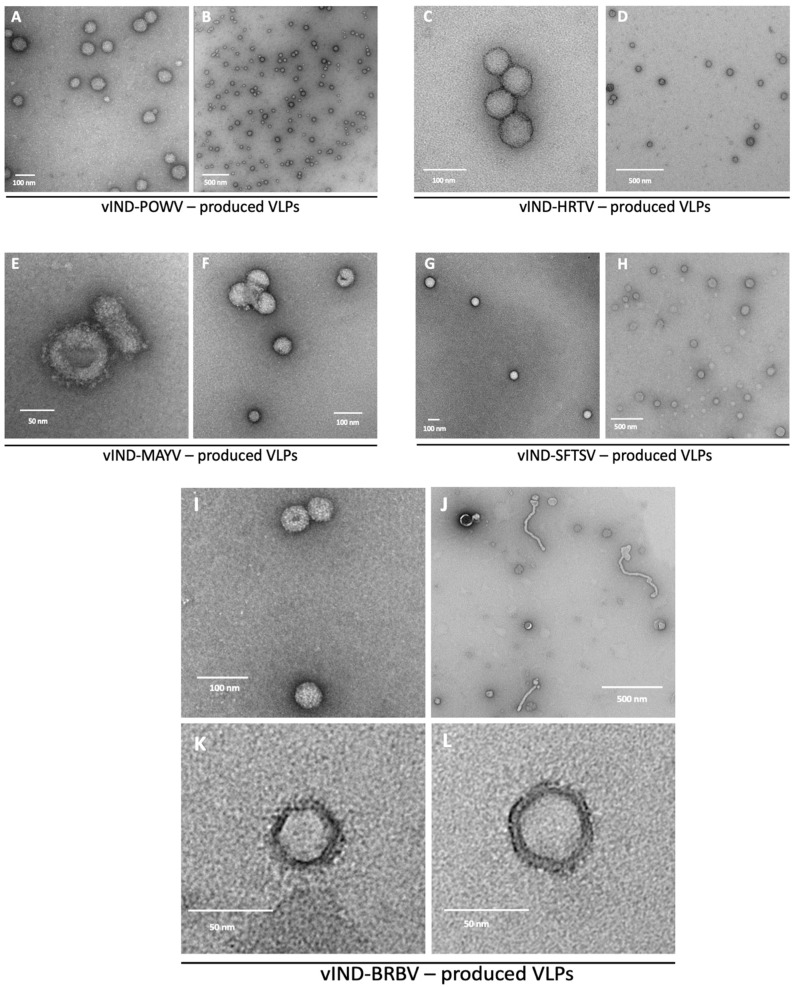
Representative TEM images of VLPs produced in Vero cells infected by vINDs in the absence of tetracyclines. (**A**,**B**) POWV (prototype lineage) VLPs. (**C**,**D**) HRTV VLPs. (**E**,**F**) MAYV VLPs. (**G**,**H**) SFTSV VLPs. (**I**–**L**) BRBV VLPs.

**Figure 4 pathogens-11-01505-f004:**
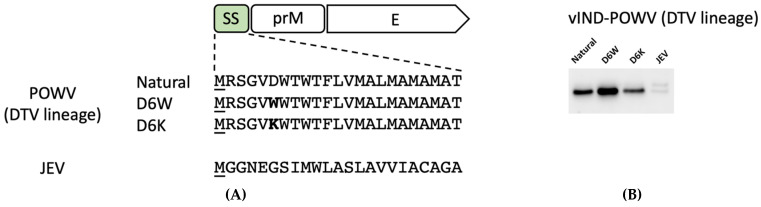
Signal sequence (SS) modifications influenced POWV VLP secretion. (**A**) Depiction of modifications in the signal sequences. Amino acid sequences are shown. A methionine (underlined M) was added to the N-terminus of each signal peptide and each mutation is bolded accordingly. (**B**) Western blots of PEG-precipitated POWV VLPs produced in Vero cells in the absence of tetracyclines. A major band of 54 kDa and a minor band (not always detectable) of 60 kDa were observed. Full-length blots with negative controls are provided in Appendix A.

**Table 1 pathogens-11-01505-t001:** Efficacy of signal sequence (SS) modification on POWV (DTV lineage) VLP secretion.

SS Modification	Prediction Score	Observed Change in Secretion Efficacy *
Natural	0.886	N/A
D6W	0.905	**↑**
D6K	0.850	=
JEV	0.886	**↓**

* ‘N/A’: not applicable; ‘=’: similar; ‘↑’: increased; ‘↓’: decreased.

## Data Availability

All data generated or analyzed during the current study are available from the corresponding author on reasonable request.

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
