# Peer review of "Generation of Multiple Arbovirus-like Particles Using a Rapid Recombinant Vaccinia Virus Expression Platform"

_pathogens, 2022, doi:10.3390/pathogens11121505_

Round 1
Reviewer 1 Report
Wang et al. describe further usage of their vector vaccine platform. The manuscript is well written and comprehensible. The authors should check the manuscript for consistency.
As they claim a use as potential vaccine in future it would be interesting to see the immunogenicity demonstrated in an infection model if applicable.
Author Response
Our previous study (reference 30 in this manuscript) demonstrated that vIND-ZIKV, a vaccine candidate for Zika virus, induced potent antibody and CMI immune responses in mice, and protected from weight loss and viremia. This finding indicates the antigen of interest (ZIKV envelope protein) was sufficiently expressed in vivo in the absence of doxycycline to elicit protective immune responses. A statement in this regard was added at the end of the Discussion.Reviewer 2 Report
Major comments:
Line 137: In the VLPs production and purification section: Why only BRBV and MAYV, but not POWV, HRTV, and SFTSV, VLPs needed further purified through sucrose cushion and linear sucrose gradient.
Minor comments
1. Line 155: “conju- gated with HRT” à “conju- gated with HRP”
2. Line 295 “ebasence” à “absence”
3. Line 406: The reference is incomplete.
4. Line 412: The reference is incomplete.
5. Line 422: The reference is incomplete.
6. Line 424: The reference is incomplete.
7. Line 437: The reference is incomplete.
8. Line 476: The reference is incomplete.
9. Line 541: The reference is incomplete.
10. Line 588: The reference is incomplete.
Author Response
Response to the major comments:
According to our practices and observation, VLPs of POWV, HRTV, and SFTSV were often sufficiently identifiable upon PEG precipitation when they were imaged by electron microscopy. BRBV and MAYV VLPs were not equally identifiable after PEG precipitation only. Further purification using sucrose cushion and gradient was necessary to obtain clear TEM images of BRBV and MAYV VLPs. In fact, for each VLP sample, we saved aliquots after each step (PEG precipitation, sucrose cushion, and sucrose gradient) and imaged them by TEM individually. Clear images were obtained for POWV, HRTV, and SFTSV VLPs that were only PEG precipitated. However, for BRBV and MAYV VLPs, they were only clearly visible after sucrose gradient purification. Clarifying statements were added to the Materials and Methods and Results section. Response to the minor comments: Amended in situ as noted.Reviewer 3 Report
The authors of “Generation of Multiple Arbovirus-like Particles Using a Rapid Recombinant Vaccinia Virus Expression Platform" have engineered a clever system to eliminate parental strains during recombinant VV production. They now have used it to produce recombinant VV expressing arboviral proteins that might auto-assemble in VLP, while preventing VV replication within the host, which is interesting for safety reasons.
This platform can thus be used for efficient generation of safe recombinant vaccines to fight emerging arboviral diseases.
However, some points remain to be clarified:
Major comments:
- VLP purification: In M&M (lines 137-139), the authors indicate that only BRBV and MAYV VLPs were purified on sucrose cushion and gradient. Yet, in the result section (lines 251-253), they write: “To image VLPs under TEM, we further purified PEG-precipitated VLPs via sucrose cushion followed by sucrose gradient.” without specifying. This needs to be clarified.
- The authors say that the gene of interest (GOI), under the control of an early/late promoter, will be transcribed even in the absence of DOX. However, the vaccinia virus early transcription factor (VETF) is a composed of 2 subunits encoded by the D6R and A8L genes, and without DOX, D6R is not expressed, and one can suppose that without replication, the late transcription factors will not be available. Do the author know if another TF can compensate for the absence of D6R to express the GOI in the absence of DOX?
- Figure 2B. Negative controls are not included the figure (they appear only in suppl figures 1S and 2S), which would be fine if they were completely negative. However, they are not blank for SFTSV nor BRBV.
Regarding BRBV, there is no mention in the text of the fact that 2 bands can be seen in the control at the same place as for the viral proteins. Of course, the intensity of both bands suggests it can be a spillover from the other lane. Yet I think this should be mentioned somewhere.
Regarding SFTSV, the intensities of both 50kDa bands are similar, and this fact can only be seen in the suppl data. I acknowledge the fact that it is mentioned in the discussion, however, I strongly recommend to include all negative controls in fig 2B for clarity.
- In the legend of figure 4, the authors indicate that WB are made with VLPs produced in Vero cells in the absence of tetracyclines. Is it the case for all WB and for TEM images? This could be helpful to the readers to specify this (as well as in the 2.5 paragraph in M&M) for each image, since it is indeed a major result if VLPs are obtained in the absence of DOX.
- If indeed VLPs are produced in the supernatant of Vero cells cultures in the absence of Tetracyclines, are the authors sure that this will be the case when vINDs are used as vaccines? Indeed, the vINDs will disseminate in the body, and in the absence of replication (which is very good news!), will there be enough produced antigen to allow the assembly of VLPs? Maybe the authors would need to be a little more cautious in the discussion, unless they have in vivo data to support this.
Minor comments:
Line 171: write within instead of wihin
Line 187: According to ref 37, Psel is an E/L promoter and note a late promoter
Figure 3: letters A to H are missing on the images
Line 295: write absence instead of ebasence
Author Response
- To the first comment:
According to our practices and observation, VLPs of POWV, HRTV, and SFTSV were often sufficiently identifiable upon PEG precipitation when they were imaged by electron microscopy. BRBV and MAYV VLPs were not equally identifiable after PEG precipitation only. Further purification using sucrose cushion and gradient was necessary to obtain clear TEM images of BRBV and MAYV VLPs. In fact, for each VLP sample, we saved aliquots after each step (PEG precipitation, sucrose cushion, and sucrose gradient) and imaged them by TEM individually. Clear images were obtained for POWV, HRTV, and SFTSV VLPs that were only PEG precipitated. However, for BRBV and MAYV VLPs, they were only clearly visible after sucrose gradient purification. Clarifying statements were added to the Materials and Methods and Results section.
- To the second comment:
A previous study of ours (reference 34 in this manuscript) took an initial step toward designing replication-inducible VACVs and showed that, in the absence of doxycycline (i.e., no active replication due to the lack of D6R expression), the gene of interest (EGFP) under the control of PE/L was still highly expressed, which was seen as green individually infected cells whose neighboring cells did not show cytopathic effects or expression of EGFP. This indicates that gene expression from the PE/L promoter was not compromised in the absence of DOX, consistent with previous reports that newly synthetizedVETF is not required for early or late gene expression [1,2 below]. In other words, VETF brought inside vIND virions (produced in the presence of tetracyclines) are sufficient to induce intermediate and then late transcription factors. Such observation was also confirmed in two more of our studies (references 30 and 31 in this manuscript) where the genes of interest were successfully expressed and detected in the absence of doxycycline inducer. The Discussion session was edited to highlight the fact that vINDs allow high levels of expression of inserted genes of interest even in the absence of inducers.
[1] Hu, X., Carroll, L.J., Wolffe, E.J., Moss, B., 1996. De novo synthesis of the early transcription factor 70-kilodalton subunit is required for morphogenesis of vaccinia virions. Journal of Virology 70 (11), 7669–7677.
[2] Hu, X., Wolffe, E.J., Weisberg, A.S., Carroll, L.J., Moss, B., 1998. Repression of the A8L gene, encoding the early transcription factor 82-kilodalton subunit, inhibits morphogenesis of vaccinia virions. Journal of Virology 72 (1), 104–112.
- To the third comment:
The comments and suggestions are greatly appreciated. Figure 2B was updated to include negative controls for all samples. The suggestion of spillover for BRBV was included in the figure legend.
- To the fourth comment:
All VLP samples used for WB and TEM in the body of the article were produced in the absence of tetracyclines. A clarification was also added to the 2.5 paragraph in M&M and figure 3 legend.
- To the fifth comment:
We do not have an approach to quantitating antigen expression or the assembly of VLPs in vivo. However, our previous study (reference 30 in this manuscript) demonstrated that vIND-ZIKV, a vaccine candidate for Zika virus, induced potent antibody and CMI immune responses in mice, and protected from weight loss and viremia. This finding indicates the antigen of interest (ZIKV envelope protein) was sufficiently expressed in vivo in the absence of doxycycline to elicit protective immune responses. A statement in this regard was added at the end of the Discussion.
- To minor comments:
Amended in situ as noted.